# Smartphone-Based Social Distance Detection Technology with Near-Ultrasonic Signal

**DOI:** 10.3390/s22197345

**Published:** 2022-09-27

**Authors:** Naizheng Jia, Haoran Shu, Xinheng Wang, Bowen Xu, Yuzhang Xi, Can Xue, Youming Liu, Zhi Wang

**Affiliations:** 1College of Control Science and Engineering, Zhejiang University, Hangzhou 310000, China; 2School of Advanced Technology, Xi’an Jiaotong-Liverpool University, Suzhou 215123, China

**Keywords:** social distance detection, acoustic signal, mutual ranging, near-ultrasonic signal, multiple access protocol

## Abstract

With the emergence of COVID-19, social distancing detection is a crucial technique for epidemic prevention and control. However, the current mainstream detection technology cannot obtain accurate social distance in real-time. To address this problem, this paper presents a first study on smartphone-based social distance detection technology based on near-ultrasonic signals. Firstly, according to auditory characteristics of the human ear and smartphone frequency response characteristics, a group of 18 kHz–23 kHz inaudible Chirp signals accompanied with single frequency signals are designed to complete ranging and ID identification in a short time. Secondly, an improved mutual ranging algorithm is proposed by combining the cubic spline interpolation and a two-stage search to obtain robust mutual ranging performance against multipath and NLoS affect. Thirdly, a hybrid channel access protocol is proposed consisting of Chirp BOK, FDMA, and CSMA/CA to increase the number of concurrencies and reduce the probability of collision. The results show that in our ranging algorithm, 95% of the mutual ranging error within 5 m is less than 10 cm and gets the best performance compared to the other traditional methods in both LoS and NLoS. The protocol can efficiently utilize the limited near-ultrasonic channel resources and achieve a high refresh rate ranging under the premise of reducing the collision probability. Our study can realize high-precision, high-refresh-rate social distance detection on smartphones and has significant application value during an epidemic.

## 1. Introduction

Recently, the outbreak of novel coronavirus pneumonia (COVID-19) is a “black swan” event facing all mankind. The world’s politics, economy, and culture have undergone tremendous changes. In the process of fighting against the virus, people gradually realized that due to an epidemic that will not disappear quickly, it is necessary to establish a long-term prevention and control mechanism for such infectious diseases [1].

In response to a sudden epidemic, a much-debated question is whether the regular epidemic prevention and control of governments tend to go to two extremes: One is to allow the epidemic to rage for economic development, and the other is to prohibit all social activities to prevent the spread of the epidemic. Both of them have not dealt with the unbalanced relationship between epidemic prevention and economic development [2]. Therefore, using techniques to strictly control resident social safety distance and locking contact is critical in ensuring normal social operation and reducing the cost of epidemic prevention, rather than banning all social activities [3,4,5,6].

The accurate measurement of social distance is a basic technology throughout the prevention, investigation, and research of the epidemic [7]. In the aspect of preventing epidemics, accurate social distance detection in real-time and warnings is the key to controlling the spread of the epidemic. According to the US Centers for Disease Control and Prevention (CDC) research, the probability of infection is high when the distance between individuals in a population is less than 6 feet and the duration is more than 10 min [8]. Thereby, all employees are required by CDC to maintain a distance of at least “6 feet” from customers and pedestrians who are not family members [9]. Furthermore, China’s National Health Commission requires the use of “1-m lines” to limit social distancing in public places [10]. Over 100 cities in China have adopted health QR codes to facilitate the control of the novel coronavirus and work resumption. However, the QR code can only determine the location of the person scanning the code, and cannot dynamically obtain social distance. Social distancing creates huge error over time.

In recent years, with the popularity of smartphones, it is undoubtable mobile phones are the most convenient to measure social distance in real-time. Global Navigation Satellite System (GNSS) based on smartphones locates multiple users through satellites and other equipment to obtain the relative distance and overlapping area of multiple users [11], which provides meter-level accuracy. However, in the indoor environment, the electromagnetic signal is severely attenuated by the occlusion of the building, resulting in the failure of the positioning function of the GNSS-based positioning and navigation software in large indoor public places. Furthermore, GNSS RTK obtains a position accuracy of 1–5 cm [12], but RTK is not compatible with civilian smartphones and is stuck in achieving high positioning accuracy when there is occlusion or semi-occlusion between the satellite and the ground. Therefore, social distance detection based on GNSS is still limited.

To address this issue, there are a large number of techniques to realize social distance detection that can be used in smartphones, such as Wi-Fi, Bluetooth (BLE), Ultra-wideband (UWB), and Pedestrian Dead-Reckoning (PDR) [13]. However, regardless of the result that they can provide highly accurate indoor location measurement, the above-mentioned technologies still have bottlenecks in terms of accuracy, cost, and compatibility of application, respectively (Section 2).

The above technologies use electromagnetic waves for ranging and require high-precision clock synchronization, which cannot be achieved by smartphones. Thus, the use of acoustic signals for social distancing detection has certain development potential. The acoustic signal is a mechanical wave compatible with consumer-grade mobile phones, which need low requirements for time synchronization. Microsoft BeepBeep [14] proposed a mutual-ranging scheme between mobile phone nodes at an early stage, which achieves an error of ~2 cm. Figure 1 illustrates the structure of BeepBeep. However, the signal of it is audible sound (4 kHz–6 kHz), which brings noise pollution to guests, and the system has no detailed investigation of NLoS and the refresh rate. Furthermore, although BeepBeep designed a simple multiple access protocol by starting recording with a microphone and calculating a proper delay, this method needs too long a signal ranging period and lacks experiment and simulation analysis. Therefore, it is not appropriate for social distance detection.

In response to the challenges and problems, we first suppose a novel acoustic social distance detection system based on mobile phones, and its performance is evaluated. Hence, our main contributions are:Firstly, a high-precision smartphone-based social distance detection technology with a near-ultrasonic signal is proposed;Secondly, combined with short-distance crowd channel characteristics, a group of 18 kHz–23 kHz Chirp signals with single frequency signals are designed and optimized to support ranging and coding;Thirdly, a precise mutual ranging algorithm is performed by using the cubic spline interpolation and two-stage search to obtain the robust mutual ranging results against multipath and NLoS affect. Both realize social distance detection with a high refresh rate and high accuracy;Additionally, combined with Chirp BOK, FDMA, and CSMA/CA, a hybrid channel access protocol is proposed. The simulation based on measured parameters verifies the protocol can increase the number of concurrencies and reduce the probability of collision;Furthermore, a considerable amount of experiments of several scenarios are carried out and demonstrate the robustness and the feasibility of this system;The system architecture of the system involved in this article is shown in Figure 2.

## 2. Related Works

Nevertheless, there are several technologies supporting positioning and ranging based on smartphones, but they remain challenging in the application of smartphone mutual ranging.

### 2.1. Wi-Fi

Among them, Wi-Fi based on Wi-Fi RSSI [15], fingerprint [16], and Wi-Fi RTT [17] are compatible with smartphones and require no additional custom hardware. Google in Android 9 was able to reach an accuracy of 1 m-2 m based on a new API. Whereas all of them require special Wi-Fi access points (AP), which causes impossibility to interact ranging between smartphones. Wi-Fi RSSI was carried out earlier, but it is difficult to accurately estimate the channel attenuation model due to the complex indoor environment and the serious NLoS, which will affect the positioning accuracy. Wi-Fi AOA and Wi-Fi CSI are also applicable to high-precision distance measuring, but they are incompatible with smartphones [18,19].

### 2.2. BLE

BLE RSSI obtains better performance than Wi-Fi RSSI. Google and Apple [20] use the interface to measure social distance using Bluetooth RSSI, and the government of Singapore launched the Bluetooth-based Blue Trace project [21]. To assist coronavirus contact tracing, Ref. [22] made a report about BLE received signal strength in different scenarios, which provides a novel application. However, it also finds that the relationship between BLE strength and transmission distance is not unique, which cause a challenge in contact tracing. BLE can also utilize fingerprints to realize a positioning accuracy of 4 m [23]. BLE AOA/AOD promotes the prosperity of indoor positioning with centimetre-level positioning accuracy [24]. Besides, IBeacon system introduced by Apple Company is based on the RSSI ranging method, and the positioning accuracy can reach 2–3 m [25]. Quuppa offers a one-size-fits-all technology solution for tracking tags and devices in real time with centimetre-level accuracy [26]. However, due to the short signal transmission distance, it is necessary to deploy many base stations (BS) to achieve high-precision positioning. BLE AOA is expensive to produce. IBeacon provides a novel smartphone mutual ranging method, but because RSSI cannot be applied in long distances, it is only suitable for the ranging range of 1 m~2 m. Therefore, BLE is limited by its characteristics.

### 2.3. UWB

UWB can also achieve ranging accuracy of 5 cm to 10 cm in ideal scenarios [27] based on RSSI [28], TDOA [29,30], and TOA [31]. UWB ToF ranging can obtain reliable dm-level accuracy. UWB systems are also generally resistant to multipath interference [32], but only a few mobile phones, such as Samsung Galaxy Note 20 or Apple iPhone 11, have UWB modules, which can cause additional energy consumption [33]. Furthermore, UWB only supports the interaction between the mobile phones and the tags rather than the interaction between phones and phones. Hence, it is also challenging to utilize UWB modules in smartphone-based social distance detection.

### 2.4. PDR

PDR mainly uses the accelerometer to measure the speed and then uses the magnetometer and gyroscope to determine the heading, so as to calculate the relative displacement of the pedestrian [34,35]. However, due to the serious electromagnetic interference in the environment, it is difficult for PDR to accurately estimate the heading angle, resulting in increased positioning errors. Hence, UWB and PDR cannot be applied to smartphone-based social distance detection.

### 2.5. Acoustic

Many acoustic ranging and positioning systems have been developed recently; Refs. [36,37,38,39] combined the indoor positioning and tracking system of the acoustic signal custom equipment to obtain the location information of the target. However, this method usually requires additional equipment such as acoustic BSs, which results in a limit in social distance detection. To realize robust ranging, Refs. [40,41] utilize the Chirp signal, but the NLOS effect is not considered in the algorithm experiment. In [42], fractional Fourier transform is applied to avoid the multipath effect, but the complexity of it is overwhelmed. A novel encoding and distance detection system by Chirp is shown by [43], but the signal length is too long (300 ms), causing a low-ranging refresh rate. Although Ref. [44] is one of the few studies using acoustic signals to detect social distance, the precision and robustness are not satisfactory.

When implementing distance detection systems, ranging accuracy, range, cost, and smartphone compatibility are considered as four principal factors. The comparison is presented in Table 1. In Table 1, the acoustic signal is suitable for social distance detection based on smartphones.

In summary, none of the technologies mentioned can well achieve mutual ranging between mobile phones, thereby detecting social distance easily and robustly.

## 3. Framework of the System

### 3.1. Mutual Ranging Principal of BeepBeep

If the clocks between the two nodes can be synchronized accurately, the distance between the two nodes can be calculated by the time of arrival (TOA). However, due to the calling synchronization error of the operating system, it is difficult to achieve real-time synchronization. To address this situation, mutual ranging can be applied. Figure 3 shows the process of smartphone mutual ranging [14].

Firstly, the time difference $T_{p}$ is presented as:(1)$$T_{p}=\frac{1}{2}(T_{S}-T_{R}),$$
where $T_{S}$ and $T_{R}$ are the time of arrival between node A and node B, and the time of arrival between node B and node A, respectively.

The final distance between smartphones is shown as:(2)$$D_{BC}=\frac{1}{2}v_{s}(T_{S}-T_{R}),$$
where $v_{s}$ is the speed of sound.

Due to the delay in the processing of the operating system and application software, this method can calculate the distance between nodes without precise clock synchronization. Hence, it is significant to accurately measure $T_{S}$ and $T_{R}$. Above analysis (Figure 3, Equations (1) and (2)) is based on the conceptual framework proposed by BeepBeep. However, in practice, the signals sent by mobile phones are often affected by reverberation and NLoS, which makes it impossible to accurately detect the arrival time.

Meanwhile, as Figure 4 shows, there are usually *N* nodes (respectively *A*, *B*, *C*, …, *N*) in a node cluster in the broadcast mode. According to the unilateral bidirectional ranging, theoretically, each node needs to send *N* − 1 signals (such as node *A.* It is necessary to conduct interactive ranging with Node *B* to Node *N* respectively). In Equation (2), $T_{S}$ and $T_{R}$ are processed differently. Hence, although the nodes that measure each other may not receive the signal from each other immediately, it will not affect the ranging accuracy [45].

Whereas due to the inability of precise clock synchronization between nodes, Signals sent by multiple nodes are randomly sent at the receiving end, resulting in overlapping collisions, that is, multiple access interference (MAI).

In summary, the mutual ranging principle of BeepBeep is difficult to overcome the interference of ranging error and MAI. Therefore, it is significant to design a suitable multiple access protocol (MAC) and robust signal detection algorithms.

### 3.2. Optimized Signal Designed

#### 3.2.1. Signal Frequency Band

In social distancing detection scenarios, it is necessary to use sound signal frequency bands that cannot be heard by human ears but can be sent and received by mobile phones. First of all, this study uses a standard sound source to test the microphone performance of a variety of mobile phones in an anechoic chamber. Figure 5 shows the frequency responses of different commercial mobile phone microphones. In order to avoid interference from ambient noise, we conduct the test in an anechoic chamber. Use the sound source to emit white noise in the full frequency band. In order to avoid the interference of the frequency response characteristics of the sound source on the test results, we use a microphone with relatively balanced frequency response characteristics to record audio at the same time as the device under test to calibrate the frequency response characteristics. In an experiment, the microphone and the device under test record the acoustic signal for 20 s at the same time, and obtain the corresponding frequency response curve. The measurement result is obtained by subtracting the frequency response curve of the standard microphone from the frequency response curve of the device under test. It can be found that the signal strength of different frequency signals recorded by the mobile phone is different, and there is a certain decline in the higher frequency band. The frequency band available in mobile phones is 0 Hz–23 kHz.

Notably, the signal frequency band should make the signal inaudible. According to the research on human hearing, the human ear has different sensitivity to sound signals from 20 Hz to 20 kHz. Under the same sound pressure level, it is most sensitive to the sound signal of 1 kHz–5 kHz, while the sound signal above 18 kHz is not. It can be heard by the human ear without causing additional noise pollution. Figure 6 presents the relationship between the hearing threshold of the human ear and the average sound pressure level (SPL) of a linear frequency modulation (LFM) signal with 18 kHz and 23 kHz in the above frequency bands as a function of distance. The LFM signal is used to cover the full frequency band. The theory relationship between hearing threshold and frequency is shown as [46]:(3)$$Tq(f)=3.64\times {\left(\frac{f}{1000}\right)}^{-0.8}-6.5\times e^{-0.6\times {\left(\frac{f}{1000}-3.3\right)}^{2}}+10^{-3}\times {\left(\frac{f}{1000}\right)}^{4}$$

It can be found that the maximum SPL does not exceed the human hearing threshold of 18 kHz. Figure 6b illustrates that the sound pressure level of near ultrasonic signal attenuates rapidly at long distance. Two hop or more apart node clusters can perform mutual ranging at the same time without long-distance conflict. Therefore, the frequency band can be used in social distance detection.

According to the frequency response curve of the mobile phone and the hearing threshold of the human ear, this paper uses the acoustic signal frequency range of 18 kHz–23 kHz for social distance detection, which is called near-ultrasonic signal.

#### 3.2.2. Signal Wave Designed—Chirp Signal

Since the acoustic signal will be affected by the multipath effect, environment noise and the Doppler effect, which makes traditional single-frequency coded signals difficult to use [47]. To deal with the problem, a well-compressed chirp signal is used. Here, Chirp is defined as:(4)$$x(t)=\cos\left(2\pi \left(f_{0}t+\frac{1}{2}kt^{2}\right)\right),\;$$
where $f_{0}$ is the starting frequency (carrier frequency), $f_{1}$ is the end frequency, *k* is the frequency modulation slope and $k=\frac{f_{1}-f_{0}}{T}$, and *T* is the signal time. For Up-Chirp, *k* > 0 and *k* < 0 for Down-Chirp. Figure 7 shows the Up-Chirp with a frequency range of 17 kHz–18 kHz. Here, the sampling rate is set to 48 kHz based on smartphone devices.

However, when the speaker sends out near ultrasonic band signals, audible noise will appear at the beginning and end of the sound, which is called low-frequency leakage. In order to solve this problem, the waveform of the chirp signal is reconstructed, and the window of Blackman window and rectangular window is used to process the LFM signal x(n). Here, the rectangular window can save all of the signal but there is no correction capability for low frequency leakage. The Blackman window can smooth the signal at the start and end of the signal, but it will cause a loss of signal strength. Hence, using two windows can not only realize the stable change of signal amplitude but also retain the energy intensity of the middle part of the signal to ensure the transmission distance.
(5)$$\hat{x}(n)=x(n)*w(n)$$
w(n) is shown as:(6)$$w(n)=\{\begin{array}{cc} 1, & \frac{1}{4}N\leq n\leq \frac{3}{4}N\\ 0.42-0.5\cos\left(\frac{\pi n}{N}\right)+0.08\cos\left(\frac{2\pi n}{N}\right), & \;\mathrm{otherwise}\; \end{array},$$
where N is the length of the signal.

Chirp signals can easily use the auto-correlation properties to determine the type and arrival time of the signal due to its compression. The auto-correlation can be described as:(7)$$\begin{array}{l} r(t)=\int_{-\infty }^{+\infty }x(t)\bar{x}(t-\tau )d\tau \\ \;\;\;\;\;\;=\sqrt{kT^{2}}\frac{\sin(\pi kTt(1-|t|/T))}{\pi kTt}\cos\left(2\pi f_{0}t\right),-\frac{T}{2}⩽t⩽\frac{T}{2} \end{array}$$
where *T* is the time of signal duration, and $kT^{2}=\left|(f_{1}-f_{0})T\right|$. Equation (7) demonstrates the larger $(f_{1}-f_{0})T$ and the larger r(t), which results in a smaller probability of decoding and error of ranging.

Figure 8 and Figure 9 illustrate under different SNR, the correlation value relationship between $\left(f_{1}-f_{0}\right)$ and *T*. The noise of SNR is white Gaussian noise and the additive white Gaussian noise (AWGN) channel is implemented to evaluate anti interference capability of the signal. To evaluate the quality of the cross-correlation results, the impulse response width (IRW) is introduced. A large IRW will result in lower ranging accuracy and an IRW that is too small will lead to a waste of frequency band and time resources. Figure 8 shows that in SNR of 20 dB, with a bandwidth of 1 kHz, as *T* increases, the IRW decreases steadily. When *T* = 20 ms, the IRW is 2.7 ms, and when *T* is longer than 20 ms, the IRW will be smaller than 2 ms, which will cause a waste of time. Figure 9 shows that when *T* is 20 ms, the IRW decreases as the bandwidth increases. If the bandwidth is smaller than 1 kHz, the IRW will be larger than 4 ms, which will cause the ranging accuracy to decrease. Besides, the simulation results also illustrate that the cross-correlation can well offset the interference caused by white Gaussian noise. In order to consider the ranging accuracy and refresh rate, we chose a Chirp signal of $(f_{1}-f_{0})=1000$, *T* = 20 ms.

#### 3.2.3. Sub-Band Signal Wave Designed

The frequency band available in mobile phones is 18 kHz–23 kHz, however, in the case of high refresh rates and crowded access nodes, there is a limit on the frequency band.

Chirp signal can achieve high-precision delay estimation, Chirp Binary Orthogonal Keying (BOK) is able to encode 0 and 1 by up-Chirp and down-Chirp. However, under Chirp BOK, the amount of data that can be transmitted per unit of time is small, resulting in an excessively long total signal length and a low refresh rate for multi-node ranging.

To overcome the problem, we proposed a sub-band signal that uses Chirp for ranging and ID identification (coding and encoding), and a single-frequency signal is utilized for auxiliary coding. Based on the analysis of signal waves and signal bandwidth, this paper divides 18 kHz–23 kHz into 5 sub-bands and performs channel allocation based on this. In order to complete the multi-node ranging and encoding and decoding in a short time, that is, to achieve a higher refresh rate, this paper superimposes the Chirp signal and the single-frequency signal in the time domain and completes the ranging and ID identification functions at the same time. In addition, pseudo-orthogonality exists between Chirp signals in different frequency bands, up-Chirp and down-Chirp, which is convenient for increasing the concurrency when multi-node mutual ranging. The design of the signals is shown in Figure 10.

To superimpose the two signals in the time domain to realize ranging and coding functions at the same time, the signal design needs to consider the following elements:Up-Chirp and down-Chirp can be concurrent. When the up-Chirp and the down-Chirp overlap and conflict at the receiving end, it is possible to distinguish which Chirp signal the single-frequency signal corresponds to;It still guarantees a high delay estimation resolution;Due to the possible Doppler frequency shift, a guard interval needs to be reserved between different single-frequency signals.

Take the 18 kHz–19 kHz Chirp signal as an example, there are three signals shown in Figure 11. Figure 11a,d,g represents the superposition of the Chirp signal and the 18,500 Hz signal, the original Chirp signal, and the Chirp signal and the 19,000 Hz signal, respectively. Figure 11b,e,h shows the cross-correlation results with Chirp of 18 kHz–19 kHz. Under the same frequency band as the single-frequency signal, the cross-correlation side lobes will increase, which results in a decrease in the ranging performance of the Chirp signal. However, as in Figure 11e,h, if the single-frequency signal band does not overlap the Chirp frequency band, there will be no significant impact.

In addition, due to the Doppler effect that causes signal recognition errors, a guard interval is required between single frequency signals. Take the walking speed of a natural person as an example of 146 cm/s, the frequency offset $\hat{f}$ is:(8)$$\hat{f}=\frac{\nu \pm \nu_{r}}{\nu \mp \nu_{s}}f=1.008f,$$
where *f* represents the origin signal frequency, 𝜈 is the propagation speed of the sound wave, ν_𝑟_ is the speed of the mobile receiver, and ν_𝑠_ is the movement speed of the sender.

Based on Equation (8), the guard interval of 250 Hz is enough to deal with the Doppler effect.

Based on the above analysis, the following signals are designed in this paper. In the process of mutual ranging, the signal sent by the transmitter is the Chirp signal or the time domain superposition of the Chirp signal and the single-frequency signal. Among them, the frequency band of the Chirp signal component is separated from the frequency band of the single-frequency signal component. For example, Figure 10b shows that the 18 kHz–19 kHz up-Chirp can be matched with the 19,250 Hz single-frequency signal in the 19 kHz–20 kHz frequency band, and the 19 kHz–18 kHz down-Chirp can be matched with the 19,750 Hz single frequency signal in the 19 kHz–20 kHz frequency band. Take the four kinds of signals within 18 kHz–19 kHz as an example, which are respectively allocated to four nodes in the same node cluster *C* for transmission. The reason the single-frequency signals matched with the up-Chirp and the down-Chirp are different is that when the up-Chirp and the down-Chirp overlap and conflict at the receiving end, the single-frequency signal can be distinguished according to the corresponding Chirp signal.

A total of 18 kHz–23 kHz of a 5 kHz available frequency band, with 500 Hz as the guard interval, a total of 10 kinds of single-frequency signals (respectively 18,250 Hz, 18,750 Hz, 19,250 Hz, 19,750 Hz, 20,250 Hz, 20,750 Hz, 21,250 Hz, 21,750 Hz, 22,250 Hz, 22,750 Hz), with five frequency bands the up-Chirp signal and the down-Chirp signal form a total of 20 kinds of signals, which are respectively assigned to nodes 1 to 20 in the same node cluster for transmission.

### 3.3. Robust Mutual Ranging Algorithm

Based on the orthogonality and the relativity of the Chirp signals, they are able to be utilized to make signal ranging by generalized cross correlation (GCC). However, the traditional correlation algorithm is not able to obtain a good accuracy of time delay estimation. In order to rectify the problem, as Figure 12 shows, a robust mutual ranging algorithm named as env-two-stage is introduced. Firstly, a discrete GCC function is used to roughly calculate the time delay. Secondly, a cubic spline envelope is implemented to remove small noise and environmental effects. Thirdly, a two stage search is utilized to obtain an accurate ranging result.

#### 3.3.1. Traditional Correlation Algorithm

Discrete GCC function (also called match filter) of receive signal X(i) and the reference signal Y(i) is:(9)$$R_{xy}\left(n\right)=\sum_{i=1}^{N}X\left(i\right)Y\left(i+n\right)$$

However, the computational complexity of Equation (9) is high. The frequency domain GCC is introduced to decrease the complexity, which is defined as:(10)$$R_{xy}\left(n\right)=\sum_{k=0}^{n-1}fftX\left(k\right)fftY\left(k\right)\exp\left(\frac{i2\pi \tau k}{n}\right),\;$$
where $fftX\left(k\right)=\sum_{m=0}^{n-1}X\left(m\right)\exp\left(-\frac{i2\pi km}{n}\right)$ and $fftY\left(k\right)=\sum_{m=0}^{n-1}Y\left(m\right)\exp\left(-\frac{i2\pi km}{n}\right)$. Based on Equation (10), the complexity of GCC will be decreased at $o(n\log_{10}m)$.

After GCC, the time of arrival, the Up-Chirp and the Down-Chirp can be defined and judged. The time of arrive $T_{0}$ is shown as [48]:(11)$$T_{0}=\arg\max R_{xy}\left(\tau \right)$$

However, as Figure 13 shows, $T_{0}$ will be affected and lagged by the multipath effect which is presented as:(12)$${\hat{R}}_{xy}\left(n\right)=\sum_{i=1}^{L}a_{i}R_{s}\left(t-t_{i}\right)+n(t),$$
where ${\hat{R}}_{xy}\left(n\right)$ is the GCC signal received, $R_{s}\left(t\right)$ is the origin GCC result, n(t) is the environment noise, $\alpha_{i}$ is the coefficient of the *i*th path.

#### 3.3.2. Robust Mutual Ranging Algorithm

Due to this problem, Tweet [49] used parabolic interpolation to remove the multipath signal. To find the first direct arrival signal, Ref. [40] applied a two-stage search. However, the parabolic interpolation cannot fully fit the signal information, especially in a strong multipath effect. A two-stage search will also be limited by small-scale environmental noise. Therefore, a novel robust distance detection algorithm is proposed based on improved cubic spline interpolation and coarse and fine Search.

Supposing there are *n* sample points in a receive ${\hat{R}}_{xy}\left(n\right)$, the subinterval Δ can be divided as:(13)$$\Delta :a=n_{0}<n_{1}<\cdots <n_{k}=b,\;(k\geq 2)$$
where *a* and *b* are the start point and the end of the signal.

Define $h_{i}=n_{i+1}-n_{i}$, $M(n)=R_{xy}{}^{\prime \prime }(n)$, $M_{i}=M(n_{i})$, the cubic spline interpolation is shown as:(14)$$\mu_{i}M_{i-1}+2M_{i}+\lambda_{i}M_{i+1}=d_{i},i=1,2,\cdots ,n-1,\;$$
where $\mu_{i}=\frac{h_{i-1}}{h_{i-1}+h_{i}},\lambda_{i}=1-\mu_{i}$, $d_{i}=6f\left[x_{i-1},x_{i},x_{i+1}\right]$.

The enveloped signal $y=f(x_{i})$ can be calculated by:(15)$$f\left[x_{i-1},x_{i},x_{i+1}\right]=\frac{f\left[x_{i-1},x_{i}\right]-f\left[x_{i,x_{i+1}}\right]}{x_{i-1}-x_{i+1}}\;i=0,1,2,\cdots ,n-1$$
(16)$$d_{i}=6f[x_{i-1},x_{i},x_{i+1}]$$

Compared to the traditional spline cubic fitting, the proposed method can reduce computational complexity.

After spline cubic fitting, the receive f(n) can be divided into groups corresponding to 1 s time ranges $f_{r}\left(n\right)$. First, the time of arrival from the speaker of smartphone A to the microphone of smartphone A $T_{AA}$ is shown as:(17)$$T_{AA}=\underset{n}{\arg\max(\left|f_{r}(n)\right|)}$$

Then, the time of arrival from the speaker of smartphone B to the microphone of smartphone A $T_{BA}$ can be calculated as:(18)$$T_{BA}=\max(\arg\max(\left|f_{r}(i)\right|,T_{AA}+gap+\arg\max(\left|f_{r}(j)\right|)$$
where $i=1,2,\ldots ,T_{AA}-gap,j=T_{AA}-gap,\ldots ,N$, gap = 2000.

Then, $T_{BB}$ and $T_{AB}$ can be obtained as above.

To estimate the accurate time of arrival, for example of $T_{AA}$, the fixed of it is shown as:(19)$$T_{AA}{}^{\prime }=\underset{n}{\arg\max(\left|f_{r}(n)\right|>a\cdot {\mathrm{Peak}}_{AA})},0<a<1,$$
where ${\mathrm{Peak}}_{AA}$ represents the maximum correlation value at $T_{AA}$.

Ideally, $T_{AA}{}^{\prime }$ is the real time of arrival. In practice cases, due to environmental noise and the small scale of multipath, there will be spurious peaks, which result in bigger errors. Figure 14 illustrates the case of spurious peaks. For this case, let a increase gradually by 0.01 from $a_{min}$ to 1, let TS be a set of $T_{AA}{}^{\prime }$, diffTS=TS(k+1)−TS(k),k=1,2,…,N−1. Here, *N* is the length of TS.

Based on the above description, the final time of arrival ${\hat{T}}_{AA}$ is:(20)$${\hat{T}}_{AA}=TS(\arg\min(diffTS(i)>threshold)),i=1,2,\ldots ,N-1$$

Figure 14a shows the example of the env-two-stage. Under multipath effect and environmental noise, compared to the Peak method (the error is 15 cm), the algorithm has better performance (the error is 6 cm). Figure 14b presents a case that the algorithm does not apply cubic spline interpolation, spurious peaks will always occur. It can be found that combined with cubic spline interpolation, the error will be more minor.

### 3.4. Improved Identification of ID Sub-Bands

To realize efficient ID identification of different Sub-Bands, FrFt is utilized to deal with Chirp signals. FrFt can be understood as the representation method on the fractional Fourier domain formed by the signal in the time-frequency plane after the coordinate axis is rotated counterclockwise around the origin by any angle [50]. The FrFt of signal *x*(*t*) is expressed as:(21)$$X(u)=\int_{-\infty }^{+\infty }K_{p}(t,u)x(t)dt$$

$K_{p}(t,u)$ is defined as:(22)$$K_{p}(t,u)=\{\begin{array}{cc} A_{\alpha }e^{i(t^{2}\cot\frac{\alpha }{2}-ut\csc\alpha +u^{2}\cot\frac{\alpha }{2})} & \alpha \neq n\pi \\ \delta (t-u) & \alpha =2n\pi \\ \delta (t+u) & \alpha =(2n\pm 1)\pi \end{array},$$
where $A_{\alpha }=\sqrt{(1-i\cot\alpha )/2\pi }$ [51]. As for the Chirp identification, the optimal *α* is defined as $2\mathrm{arc}\cot(-\frac{f_{0}-f_{1}}{f_{s}})/\pi$, and $f_{s}$ is the sampling rate of the system. Based on the FrFt, the designed signal with the optimal *α* can be transformed into an impulse response signal, the output is shown in Figure 15. The result of FrFt is not disturbed by single frequency signals.

The center frequency of the Chirp signal is expressed as:(23)$$f_{center}=x\csc\mathrm{arccot}(-\frac{f_{0}-f_{1}}{f_{s}})$$
where x=argmaxK(u,p). According to Equation (20), the type and frequency range of Chirp can be identified. Since the signal bandwidth is preset (1 kHz), the type of chirp signal can be identified based on the value of $f_{center}$.

After that, according to the result of FrFt, short-term Fourier transform (STFT) is used to identify single frequency signals. Figure 16 illustrates the results of FrFt and STFT under SNR of 5 dB. The different nodes can be distinguished by the proposed method.

### 3.5. Hybird Channel Access Schemes

According to Section 3.1, due to MAI, it is necessary to efficiently and fairly allocate and use the acoustic channel when measuring each other in a crowd composed of multiple nodes [52]. To solve the optimization problem, Ref. [53] selects a signal sequence, but it supposes perfect synchronization between the transmitted quadrature signals and no multipath. In actual scenarios, there are only pseudo-orthogonal near-ultrasonic signals, that is, there will still be some interference between the signals. This makes the use of near-ultrasonic mutual ranging to face an extreme conflict challenge, as shown in Figure 17, so algorithms are needed to avoid conflicts as much as possible. Here, a hybrid MAC protocol is proposed to deal with a crowd conflict situation.

CSMA/CA

The Carrier Sense Multiple Access (CSMA) method improves the ALOHA method. After monitoring the channel, CSMA can adopt three types of backoff algorithms. Due to the large difference between the signal energy sent by the node and the received signal energy in the wireless local area network, it is impossible to monitor while sending and Carrier Sense Multiple Access with Collision Detection (CSMA/CD) method cannot be used. For near-ultrasonic signals of mobile phones, duplex transmission and reception can be performed, but more time resources will be wasted due to the long propagation delay of near-ultrasonic signals. Therefore, the idea of avoiding CSMA/CA is used for channel access. In CSMA/CA mode, each node can use the Distributed Coordination Function (DCF). Each node independently competes for the channel transmission right. When a node needs to send a signal, it first randomly backs up to avoid generating Collision (unless the channel has not been used recently and the channel is idle). The node that successfully receives the frame needs to send an acknowledgment immediately, if it does not receive an acknowledgment, it doubles the number of back-off time slots and retransmits until the upper limit of the number of retransmissions is reached [54].

FDMA

Frequency division multiple access (FDMA) divides the total bandwidth into multiple orthogonal channels, and each user occupies one channel. Using the FDMA method, multiple nodes can send ranging signals at the same time, so that more ranging tasks can be completed in unit time. However, when the total bandwidth resource is limited, the more mutually orthogonal frequency bands are divided and the narrower the bandwidth of each frequency band, which will reduce the accuracy of delay estimation. Therefore, if only FDMA is used, the number of concurrent nodes and node capacity is limited.

Chirp BOK

Because of the autocorrelation and energy aggregation characteristics of the Chirp signal in its time and frequency domains, Chirp BOK is a spread spectrum communication system which utilizes up-chirp and down-Chirp signals to transmit binary data. However, only the Chirp BOK method is used for signal encoding, and the amount of data that can be transmitted per unit of time is small, resulting in an excessively long total signal length and a low refresh rate for multi-node ranging.

The 20 kinds of signals in the near-ultrasonic signal time-frequency diagram designed in this paper can uniquely encode 20 nodes in the cluster, and different signals are sent by different nodes, respectively.

Compared to the frame structure in IEEE 802.11, the signal frames for social distancing scenarios have the following differences:

(1) The frame is mainly used for encoding nodes and delay estimation, and the frame length is relatively fixed;

(2) In the social distance measurement, if a control frame is added, the length of the control frame and the frame body is similar, and the acknowledgement character (ACK) frame will significantly increase the mutual ranging period, so the social distance measurement scene does not introduce the ACK confirmation frame and control frame.

The protocol should be re-designed based on the social distance background.

The near-ultrasonic signal is used for ranging and coding. Since the length of the frame is close to the maximum signal propagation delay between nodes, a longer propagation delay will increase the probability of collision. Therefore, on the basis of CSMA/CA, drawing on the idea of P-persistent, when the channel is idle and the backoff is over, it will be sent with 100% probability, but now it will be sent with P probability;Based on the signals designed, combined with the pseudo-orthogonal characteristics of FDMA and Chirp BOK, multiple pseudo-orthogonal signals can be sent simultaneously;In the CSMA/CA method, when a node needs to send data for the first time (rather than a failed retransmission), if the channel is idle, it can be sent after waiting for Distributed Inter-frame Spacing (DIFS). However, in the perception of social safety distance, at the beginning of each round of node-cluster mutual ranging, nodes may send data centrally, so a fallback scheme is adopted for all nodes.

In a round of node mutual ranging, each node only needs to send a signal once. After each node is assigned a different signal, each node first randomly selects the backoff time. Even if the channel is idle, the nodes randomly back off for a period of time before sending. If there are N nodes in total, node  i,(i=1,2,…,N) randomly selects an integer $Z_{i}$ in the range of $\left[0,CW_{i-1}\right]$ and sets the backoff counter to $Z_{i}\times T_{slot}$, where $T_{slot}$ is the length of the time slot, and $CW_{i}$ is determined by the node density and the motion status of node *i*. The node monitors the channel for a period of interframe space (IFS). If the channel is idle (the pseudo-orthogonal signal is also considered to be idle when the channel is detected), it starts to count down the back-off counter. For an orthogonal frame, the counter is suspended, and after the frame ends, the duration of the channel IFS is monitored again. If the channel is idle (the channel is considered idle when a pseudo-orthogonal signal is detected), the timer value continues to decrease. When the counter decreases to zero, the transmission frame is sent with probability *P* and delayed to the next time slot with probability (1−*P*). If it is the latter and the channel is idle, it still sends the signal with probability *P*, delays it to the next time slot with probability (1−*P*) and repeats, or another node starts to send the signal.

### 3.6. Overview of the System

To summarize, the framework of this system is shown in Figure 18. The system is divided into transmitter and receiver. The main task of the transmitter is to assign different signals to different nodes. Firstly, select a certain frequency band, such as frequency band A (18 kHz–19 kHz), then select the upper Chirp or the lower Chirp and then select the time domain to superimpose the single frequency signal or not to superimpose the single frequency signal, where the frequency of the single frequency signal is the same as that of the upper sweep frequency Chirp or the lower frequency signal. Sweep Chirp one-to-one correspondence. Through the above process, different signals are assigned to different nodes. For example, node 1 is assigned a superimposed signal of an 18 kHz–19 kHz up- Chirp and a 19,250 Hz single-frequency signal. After dealing with low-frequency leakage, each node sends a signal in the designed channel access mode, and the signal reaches the receiving end after being affected by the environment.

Secondly, signals from multiple nodes may collide and overlap at the receiver. The receiving end first uses Fractional Fourier (FrFt) to decode multiple Chirp components in the received signal. Then, according to the decoded frequency band up-Chirp or down-sweep Chirp, find the corresponding possible single-frequency signal, if the frequency band energy of the single-frequency signal in the received signal exceeds the threshold value, then the single-frequency signal exists. The above process can decode the signals of different nodes and then estimate the time delay by the method of env-two-stage to obtain the time stamps of the signals of different nodes.

Finally, for the MAI problem, we propose a new MAC protocol that integrates CSMA/CA, FDMA, and ChirpBOK. Through this protocol, the conflict of mutual ranging between different nodes can be well avoided, the period of mutual ranging can be shortened, and the efficiency of mutual ranging can be improved.

## 4. Experiment and Simulation

### 4.1. Experiment of Ranging

In order to further verify the applicability of the proposed ranging method, the three cases of the experiment are carried out. The ranging result will be compared with the traditional method, which only uses maximum cross-correlation (named as Peak) [41,48,55,56,57], two-stage search [40] (only use two-stage search) and env-two-stage proposed. The two smartphones are Huawei Mate40 Pro and Huawei P20Pro. Each experiment is conducted for 150 runs.

Figure 19 shows the experimental scenarios, which are respectively:

Case 1 LoS (Figure 19a) Indoor, quiet, meeting room: The experimental ranging distances are 1 m, 2 m, 3 m, 4 m, and 5 m. There is no obstruction between the two smartphones, and signals are transmitted directly between smartphones. The environmental temperature is 30 °C;

Case 2 NLoS by people (Figure 19b) Indoor, quiet, corridor of nucleic acid amplification testing: The experimental ranging distance is 3 m, and a human body forms a shield at 20 cm from one of the smartphones to simulate people holding mobile phones. The environmental temperature is 30 °C;

Case 3 NLoS in canteen (Figure 19c) Indoor, noisy, canteen: The experimental ranging distance is 1.5 m and the smartphones are placed on the table, with an acrylic board used as a shield between the mobile phones to simulate the situation of placing mobile phones when people eat. The environmental temperature is 22 °C.

To facilitate the collection of experimental data, this article uses Android Studio to develop Android-side testing software for basic interface calls and system testing. At the same time, the Netty open source framework is used to develop asynchronous network applications for the test. The host computer controls multiple mobile phones to automatically conduct a large amount of data collection and a large number of ranging experiments.

Figure 20 shows the mobile phone test software interface. The SEND and RECORD buttons on the interface are used for customized acoustic signal sending and recording in signal design and testing. The CONNECTION button is used for communication between multiple mobile phones and the host computer, and the host computer can send instructions for receiving and sending sound signals to multiple mobile phones respectively.

To use the near-ultrasonic signal for time delay estimation and encoding and decoding, it is necessary to obtain the original, unprocessed acoustic signal. However, generally, when the mobile phone is used to answer the call, in order to reduce the noise of the call, the multiple microphones of the mobile phone cooperate, and the audio will be automatically processed. In this article, the AudioSource parameter is selected as UNPROCESSED, so that the original audio PCM file can be obtained in the system of Android 7 and above and then the algorithm for converting to a WAV file is added in the application, and the file header is added to the WAV file. At the same time, the SAMPLE_RATE_INHZ (sampling rate) of 48,000 Hz, AUDIO_FORMAT (number of data format) select 16BIT PCM format to obtain high-quality sound signals.

Notably, in actual situations, people could hold mobile phones in various postures. In order to verify the signal characteristics generated by different poses, an acoustic imager is utilized. The acoustic imager can display the sound field strengths of the signal by means of a heat map. The four different poses, signal cross-correlation, and imaging results are shown in Figure 21. Figure 21a–d shows that though the sound signal will be partially attenuated, the sound field strengths of the acoustic signals from the mobile phone in the four postures are concentrated at the location of the speaker, which shows that the signals from the mobile phone in the above postures can be regarded as LoS. Figure 21e–h illustrates the signal correlation performance is good under four postures. This result can also illustrate that the proposed scheme is not affected by the gesture of holding the smartphone and the pocket of the clothes. In the absence of obvious obstacles, the designed signal sent by the smartphone can be regarded as a LoS signal (Case 1).

### 4.2. MAC Simulation Settings

Based on the experimental results, the proposed hybrid MAC simulation parameters are set up as shown in Table 2. The evaluation indicators are a total period of node group mutual ranging and the number of nodes in conflict. The results will be compared with ALHOA.

## 5. Result and Discussion

### 5.1. Case 1 LoS

Figure 22 shows the signal spectrum in the time and frequency domain and cross-correlation results in Case 1 at 5 m. It can be found that at this time, the signal reverberation and multipath are not serious, the time delay is short, and the peak of cross-correlation is clear.

The ranging results in Case 1 are shown in Figure 23. The box figure illustrates that in Case 1, the error distribution produced by the env-two-stage is more concentrated, with fewer outliers and smaller errors. Notably, the negative errors are only used in the box figure to show the distribution of the errors. In the CDF plot and summary table, the error is the positive error by absolute value. Currently, since the influence of multipath is not serious, the cubic spline interpolation method has the greatest correction effect on the error.

Figure 24 shows the CDF of different methods of ranging in case 1. From the CDF results in Figure 24a, under LoS, both Peak and two-stage share a similarity of performance. Meanwhile, the env-two-stage obtains a better performance in Figure 24b. Our results demonstrated that the env-two-stage gives clearly better results than other comparison algorithms in LoS.

### 5.2. Case 2 NLoS by People

Figure 25 shows the spectral (a) and cross-correlation (b) properties of signals in case 2 at a test distance are 3 m in NLoS, where the barrier is people. Obviously, the reverberation of the signal is very serious and it is difficult to search for the correct time delay.

The experiment results in Case 2 are shown in Figure 26. As can be seen in Figure 26a, under NLoS, the error distribution of env-two-stage is closer to a normal distribution than other methods. In addition, env-two-stage can achieve a mean ranging error of 0.206 m, which is the best of the contradistinctive algorithms. The reason of it is that in the situation of body occlusion, the envelope can filter small-scale environmental noise and the two-stage can overcome the multipath effect.

### 5.3. Case 3 Canteen

In Case 3, as Figure 27a presented, the signal will be reflected by the desktop and the glass plate, which results in a poor quality of the signal. Figure 27b demonstrates that in Case 3, the first peak and the largest peak are indistinguishable from the signal after cross-correlation processing. Hence, traditional methods cannot be implemented in Case 3.

Figure 28 demonstrates that in Case 3, env-two-stage is still the best method. In Figure 28a, the env-two-stage has an excellent performance in general. In Figure 28b, although median error of the two-stage is smaller than the env-two-stage, it has more outliers. On the contrary, the traditional method Peak cannot work in Case 3.

All the experimental results are summarized in Table 3. These results are obtained in Case 1, Case 2, and Case 3. The results verify that the detection accuracy of the env-two-stage algorithm is significantly better than the traditional Peak and two-stage algorithm, especially in NLoS scenes. There may be two reasons. The first is that the envelope can remove small-scale multipath and environmental noise. The second is that the algorithm part of two-stage can find the first path in the case of severe reverberation. These can also be explained in Case 1, at the distance of 1 m, the Peak algorithm obtains a better performance due to the noise and multipath are not serious in the current situation.

### 5.4. MAC Simulation

Based on experiment results, the pure contention channel access protocol and the proposed hybrid channel access protocol are simulated, respectively. In each simulation, different numbers of nodes are randomly placed within a range of five meters in diameter. In order to simulate the difference in the estimation ability of social distance between people, the distance between nodes is greater than 0.8 m. The simulations were performed 100 times to compare the efficiency of mutual ranging and the conflict situation. The specific parameters are shown in Table 2.

Figure 29 shows the relationship between node conflict situation and personnel density in the situation of CSMA/CA. It is found that with the increase of node density, the total period of node cluster mutual ranging and the probability of conflict continuously increase. The total period of mutual ranging of more than 500 milliseconds is too long, and the refresh rate is low for people whose topology changes dynamically. Through conflict avoidance, the conflict probability is reduced, but after the node density increases, the conflict probability increases rapidly.

In the case of different numbers of nodes (node density), 100 simulations were carried out, respectively, and the node group mutual ranging period and the number of nodes colliding were obtained under the hybrid channel access method.The results are shown in Figure 30.

It is found that with the increase of node density, the total period of node cluster mutual ranging continues to increase, but since the pseudo-orthogonal signals based on FDMA and Chirp BOK can be partially concurrent, the conflict probability is zero within the population density of 0.66 people/$m^{2}$, the population density continued to increase and conflict probability remained at a very low level. At the same time, the total ranging period can meet the demand. By adopting the hybrid channel access method, time resources can be utilized more evenly. Although the signals between different frequency bands are orthogonal and can be concurrent, the signals of multiple nodes are sent in a certain period, which will affect the ranging accuracy in this period. The hybrid channel access protocol adopts the idea of CSMA/CA to control the concurrency within a certain range and to solve this problem at the same time.

The performance of the two channel access protocols is compared, as shown in Figure 31. There are three conclusions that can be summarized:As the density of nodes (personnel) increases, it takes longer for node clusters to complete a round of mutual ranging under the three channel access protocols;Compared with competitive channel access, the hybrid channel access protocol proposed in this paper greatly shortens the total period of node cluster mutual ranging and can complete the task of mutual ranging in a shorter time, which is good for dealing with crowds. The characteristics of the topology structure change over time, and this improvement is more obvious at higher node densities;Compared with the competitive channel access, the hybrid channel access also greatly reduces the collision probability under the same node density, and the advantage is more obvious when the node density is larger. Among them, under the hybrid access mode, the number of conflicting nodes within 10 nodes (corresponding to a node density of 0.51 people/m^2^) is 0. Because there are 10 pseudo-orthogonal sub-channels in total, when assigning customized signals to nodes, the pseudo-orthogonal signals (pseudo-orthogonal of signals in different frequency bands and pseudo-orthogonal between Chirp BOKs) are preferentially selected.

## 6. Conclusions

In this paper, a novel smartphone-based social distance detection technology with a near-ultrasonic signal is presented and evaluated. Through the comprehensive design of the signal and the channel access, the high-precision, high-refresh-rate social distance measurement based on smartphones is realized. The main findings of the paper are as follows:A high-precision mutual ranging and coding (ID identification) method with limited bandwidth is proposed, which improves the refresh rate and accuracy of social distance detection from the aspects of signal design and delay estimation method, respectively;A robust mutual ranging algorithm env-two-stage is designed, which combines cubic interpolation and two-stage method. Experiment results show that in the case of LoS, 95% of the mutual ranging error within 5 m is less than 10 cm. In the case of NLoS, the proposed algorithm has a better performance than contrast algorithms;A hybrid channel access protocol is analyzed, which makes full use of the pseudo-orthogonality of the signal and combines Chirp BOK, FDMA, and CSMA/CA, increasing the number of concurrencies and reducing the probability of collision. Simulation results show that the proposed hybrid channel access protocol solves the problem of high collision probability caused by long acoustic signal propagation delay, and the total ranging period and collision probability are lower than the ordinary CSMA/CA method;In conclusion, this study utilizes smartphone-based near-ultrasonic signals to realize robust social distance detection. The introduction of the system proposed can increase the accuracy of distance detection and refresh rate. We have achieved an accurate and high-refresh-rate mutual ranging system, the most robust ever reported in the literature.

However, this study only verifies the feasibility of the proposed method. In the future, further investigations are required to realize a real-time system through the PC platform, Android and IOS application and WeChat mini program. Future research should be undertaken to explore the offline social distance detection system robustness in complex scenarios, such as train cabins, supermarkets, and subway stations.

## Figures and Tables

**Figure 1 sensors-22-07345-f001:**
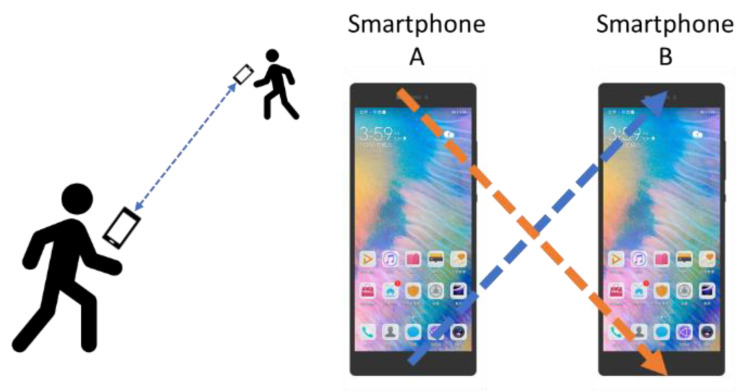
The structure of BeepBeep.

**Figure 2 sensors-22-07345-f002:**
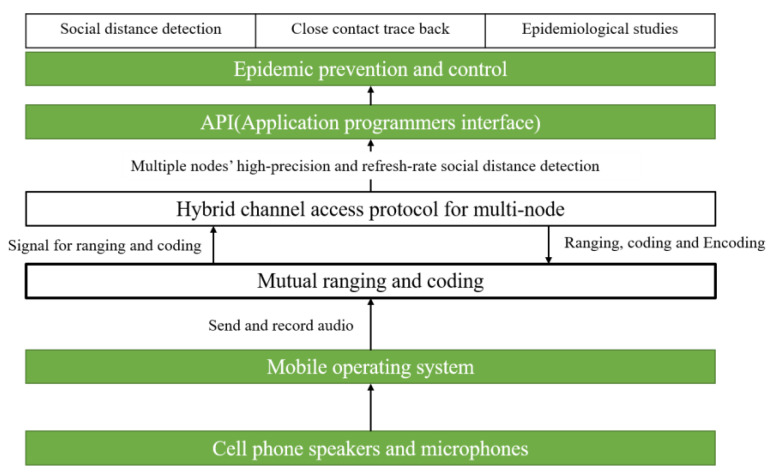
System architecture of system proposed.

**Figure 3 sensors-22-07345-f003:**
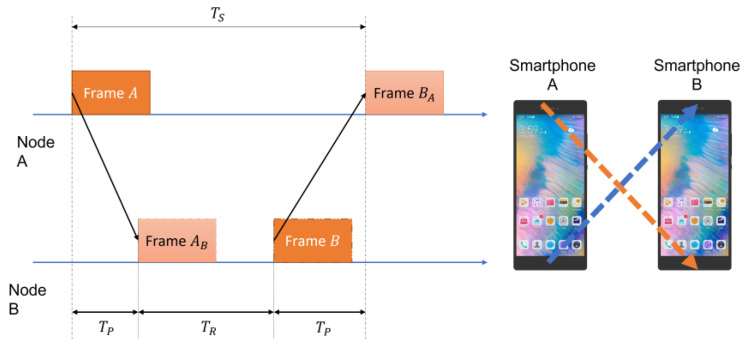
Mutual ranging flow chart based on smartphones.

**Figure 4 sensors-22-07345-f004:**
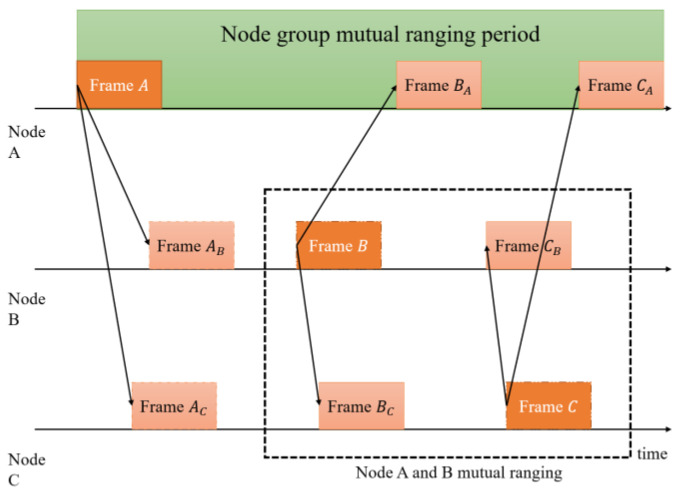
Multi-node broadcast mutual ranging.

**Figure 5 sensors-22-07345-f005:**
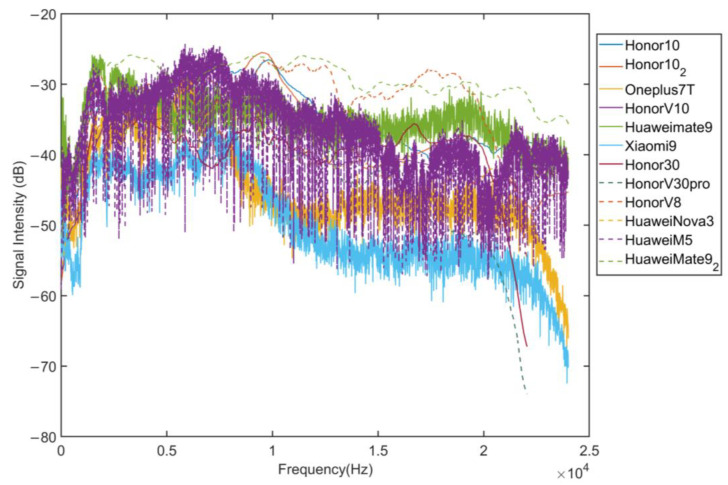
Frequency response characteristics of different mobile phones.

**Figure 6 sensors-22-07345-f006:**
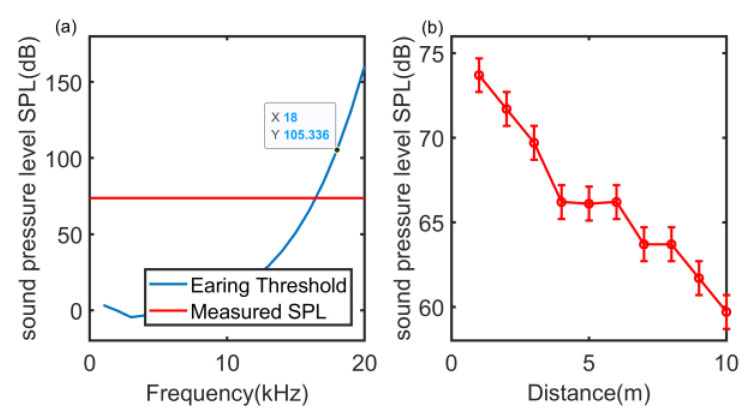
(**a**) The relationship between hearing threshold and frequency, where measured SPL represents the maximum SPL in LFM of 18 kHz–23 kHz. (**b**) The SPL with distance.

**Figure 7 sensors-22-07345-f007:**
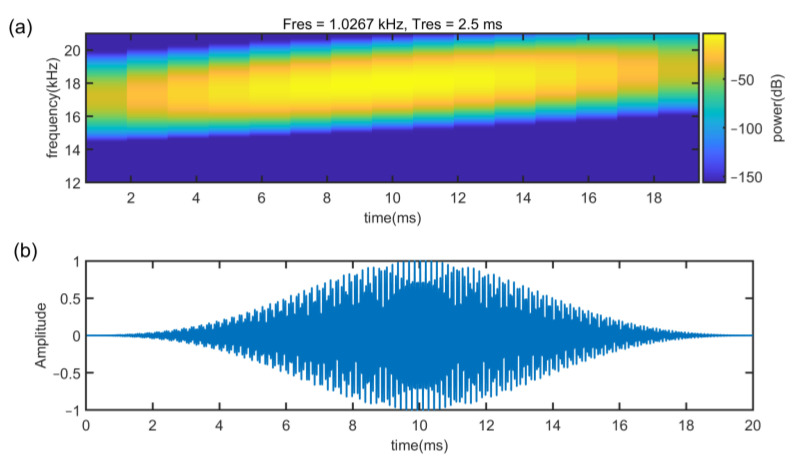
(**a**) Spectrum and (**b**) waveform of the Chirp signal.

**Figure 8 sensors-22-07345-f008:**
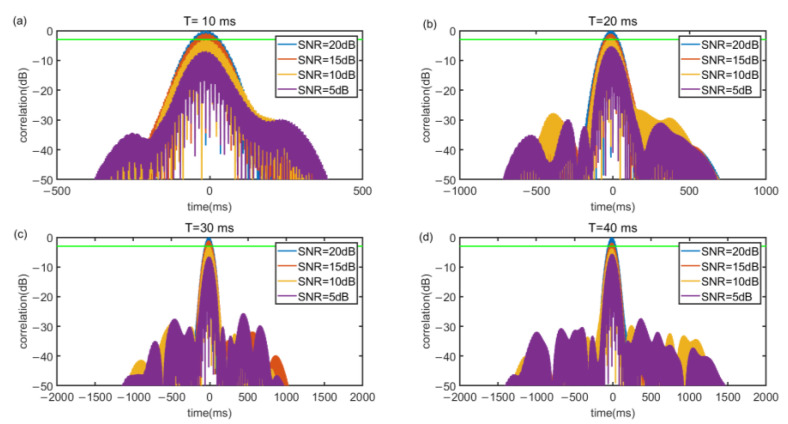
Autocorrelation power spectrograms of different *T* in the case of Gaussian noise with a bandwidth of 1 kHz. (**a**) *T* = 10 ms, IRW = 2.8 ms (**b**) *T* = 20 ms, IRW = 2.3 ms (**c**) *T* = 30 ms, IRW = 1.4 ms (**d**) *T* = 40 ms, IRW = 1.1 ms.

**Figure 9 sensors-22-07345-f009:**
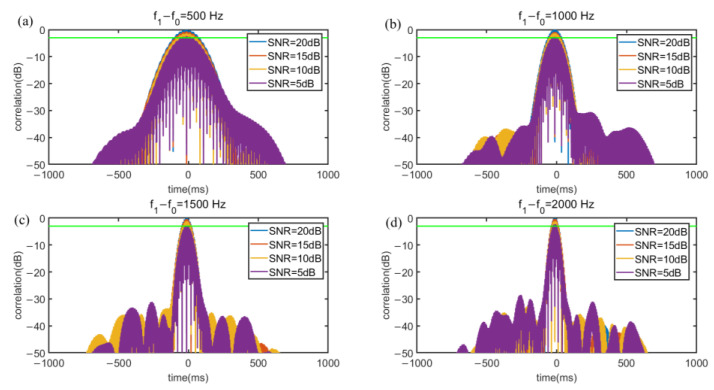
Autocorrelation power spectrograms of different bandwidths in the case of Gaussian noise with fixed *T =* 20 ms. (**a**) bandwidth = 0.5 kHz, IRW = 4.7 ms (**b**) bandwidth = 1 kHz, IRW = 2.7 ms (**c**) bandwidth= 2 kHz, IRW = 1.5 ms (**d**) bandwidth = 2.5 kHz, IRW = 1.1 ms.

**Figure 10 sensors-22-07345-f010:**
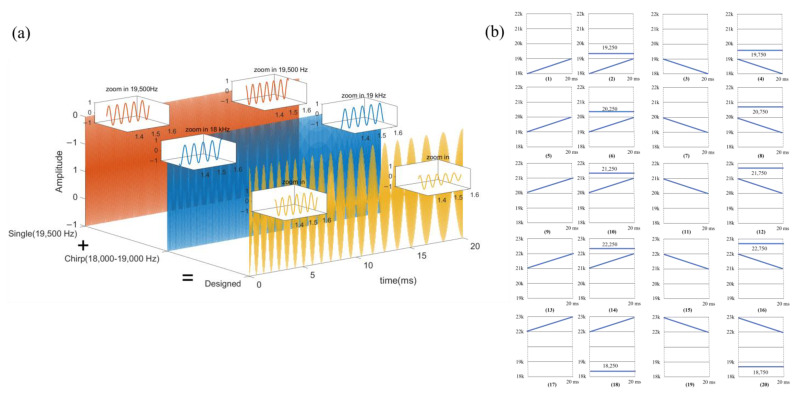
Signal design with coding and ranging functions. (**a**) Schematic diagram of the signal design. (**b**) The time-frequency diagram of the 20 designed near-ultrasonic signals.

**Figure 11 sensors-22-07345-f011:**
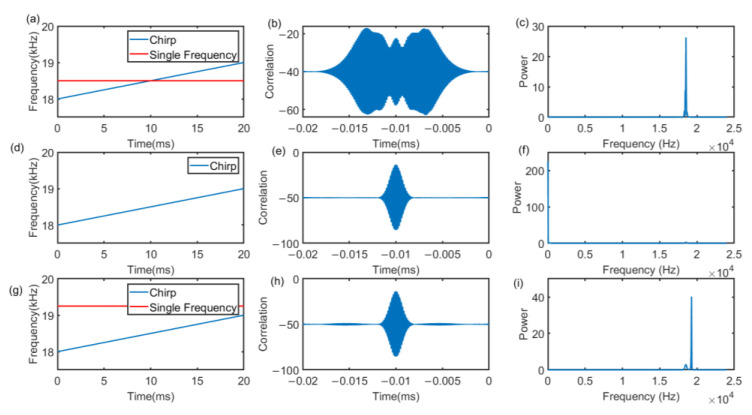
Time-frequency domain analysis of different signals. (**a**,**d**,**g**) is the time-frequency domain of the designed signal. (**b**,**e**,**h**) is the correlation results with Chirp of 18 kHz–19 kHz. (**c**,**f**,**i**) are the frequency spectrogram, respectively.

**Figure 12 sensors-22-07345-f012:**
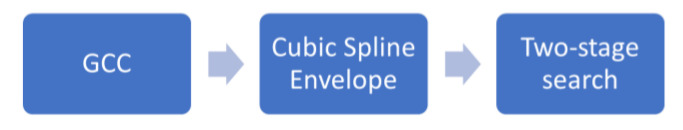
Robust Mutual Ranging Algorithm flow chart.

**Figure 13 sensors-22-07345-f013:**
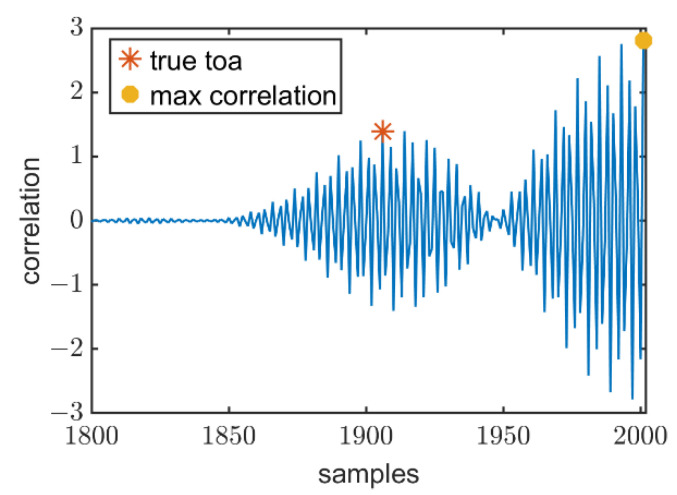
The time of arrival (TOA) is lagged due to the multipath effect.

**Figure 14 sensors-22-07345-f014:**
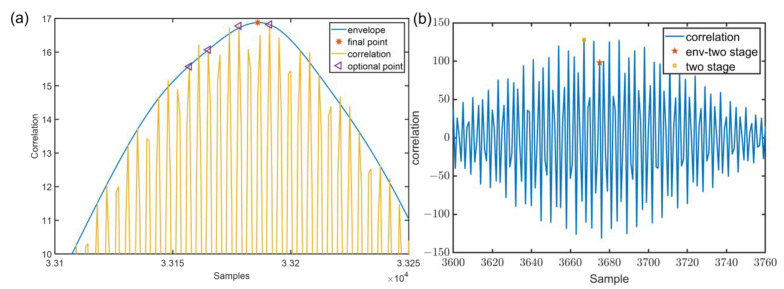
(**a**) Example of env-two-stage. (**b**) The comparison between two-stage and env-two-stage.

**Figure 15 sensors-22-07345-f015:**
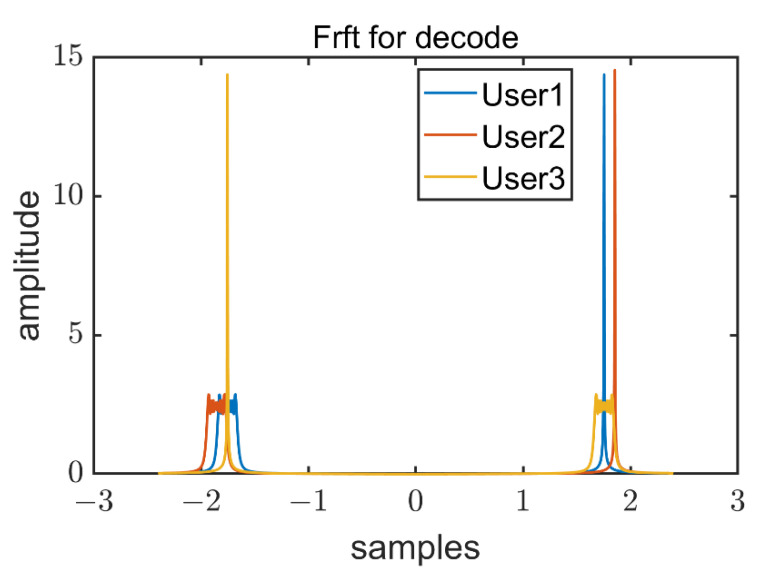
The output of FrFt with optimal α of designed signal.

**Figure 16 sensors-22-07345-f016:**
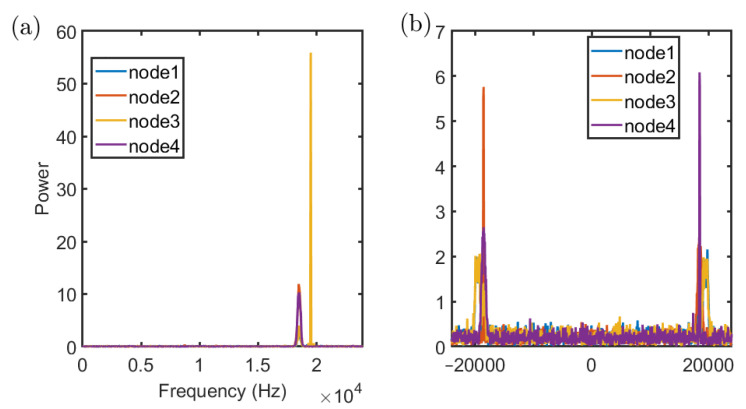
Results of STFT (**a**) and FrFt (**b**) with 4 nodes. Node 1 is an 18 kHz–19 kHz Chirp signal, superimposed with a 19,500 Hz single-frequency signal. Node 2 is an 18 kHz–19 kHz Chirp signal. Node 3 is a 19 kHz–18 kHz Chirp signal, superimposed with a 19,500 Hz single-frequency signal. Node 4 is a 19 kHz–18 kHz Chirp signal.

**Figure 17 sensors-22-07345-f017:**
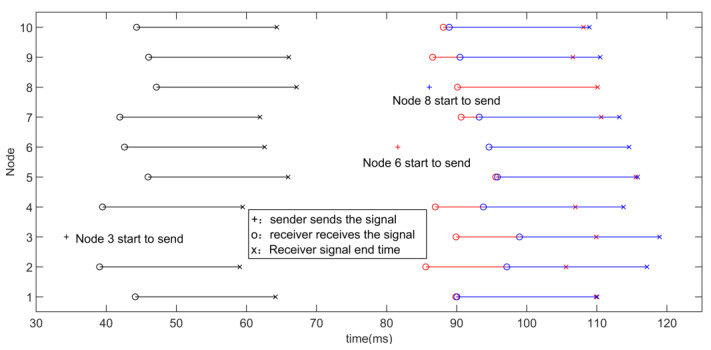
Simulation of Multi-Node Near Ultrasonic Ranging Conflict. Each row represents the time axis of each node, “+” indicates the moment when the sender sends the signal, “o” indicates the moment when the receiver receives the signal, and “x” indicates the end moment of the signal received by the receiver. Node 6 decides to send a signal at the red “+” moment, and the signal arrives at all other nodes in turn. The arrival time is marked with a red “o” in other rows (corresponding to all other nodes) and ends at a red “x”. Due to the long propagation delay, when the signal sent by node 6 has not yet reached node 8, node 8 mistakenly thinks that the channel is idle and is ready to send a signal, so it sends a signal at the blue “+” moment, and the signal sent by node 8 reaches other. For all nodes, the time of arrival is marked with a blue “o” in other rows (corresponding to all other nodes) and ends at a blue “x”. In this process, due to the long propagation delay, node 8 mistakenly thinks that the channel is idle, so it collides with the signal sent by node 6.

**Figure 18 sensors-22-07345-f018:**
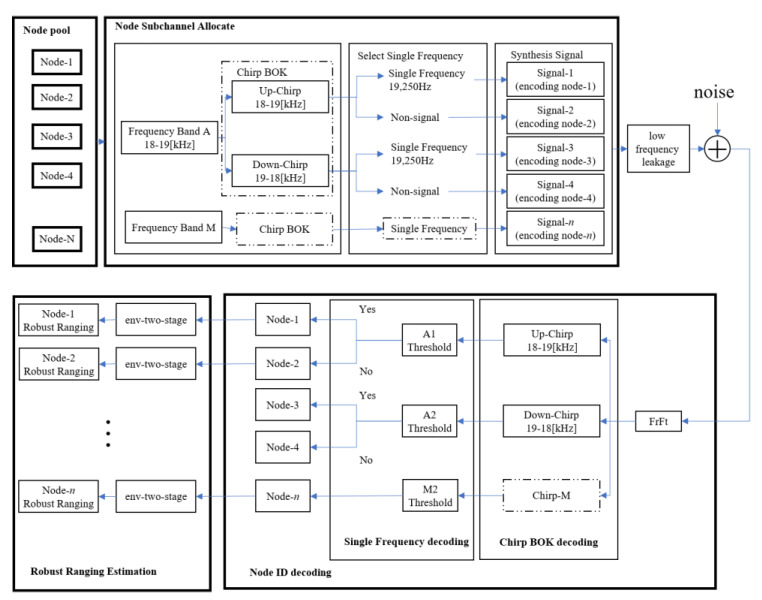
Communication system principle framework.

**Figure 19 sensors-22-07345-f019:**
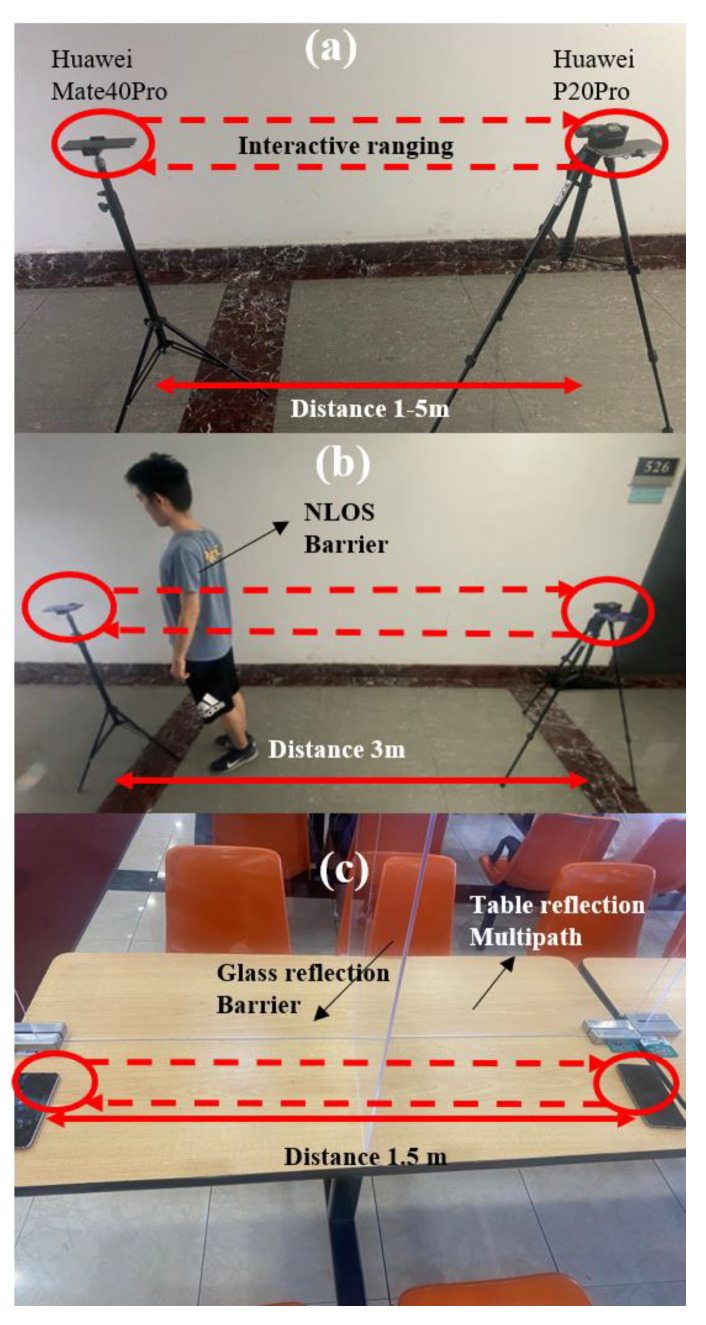
Three cases of experiments. (**a**) LoS. (**b**) NLoS, the cover is the human body. (**c**) NLoS in a canteen, the cover is glass.

**Figure 20 sensors-22-07345-f020:**
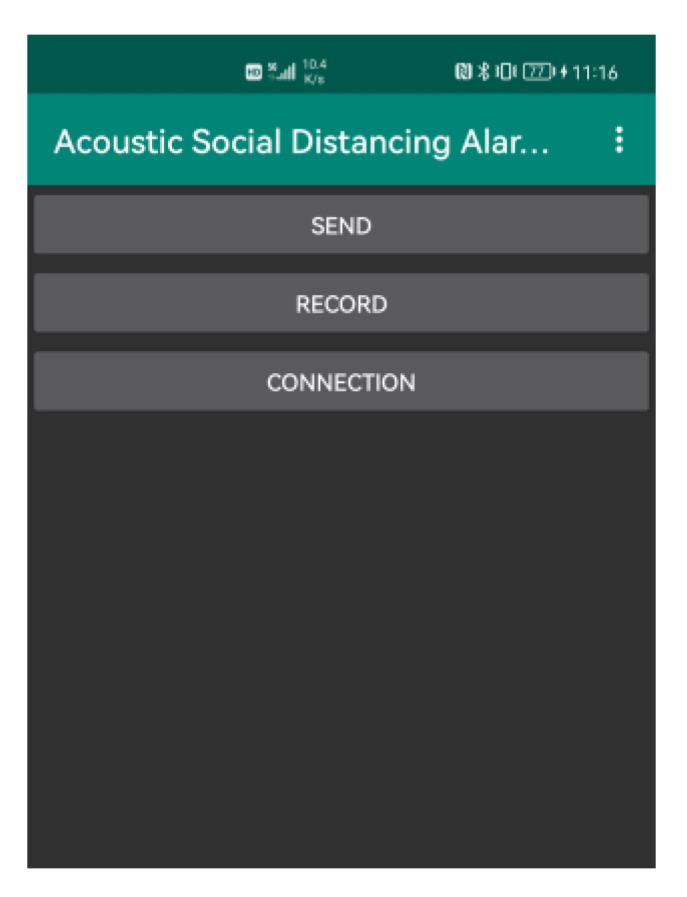
Mobile phone test software interface.

**Figure 21 sensors-22-07345-f021:**
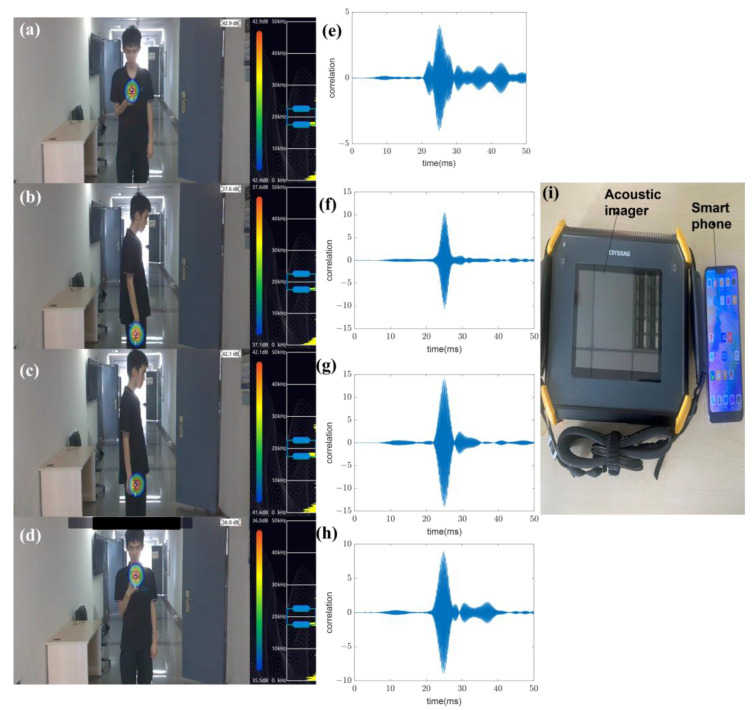
Imaging results of different poses of people holding a mobile phone. (**a**) The phone is held at an angle. (**b**) The phone sags naturally with the arm. (**c**) The phone is in a trouser pocket. (**d**) The angle of the phone is perpendicular to the ground. (**e**) correlation results of Pose (**a**). (**f**) correlation results of Pose (**b**). (**g**) correlation results of Pose (**c**). (**h**) correlation results of Pose (**d**). (**i**) The smartphone and the acoustic imager.

**Figure 22 sensors-22-07345-f022:**
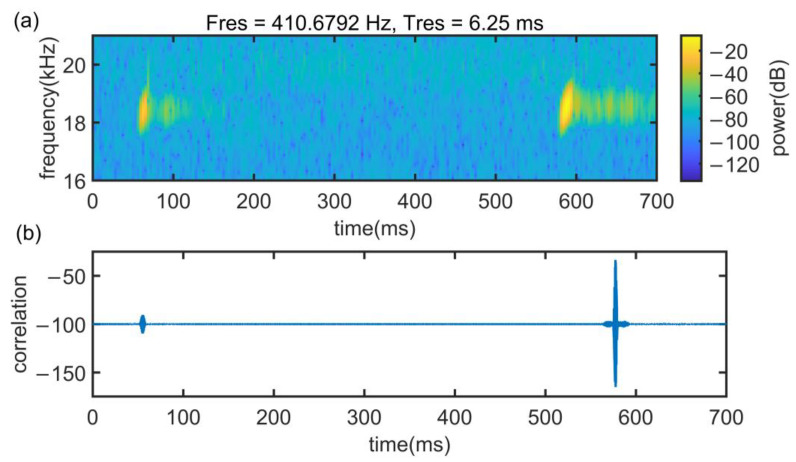
Signal propagation characteristics in case 1. (**a**) Signal spectrogram and (**b**) cross-correlation results.

**Figure 23 sensors-22-07345-f023:**
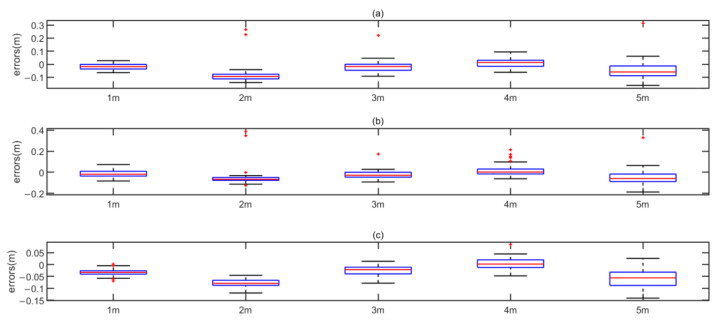
Error distribution in Case 1. (**a**) ranging in method Peak. (**b**) ranging in method two-stage. (**c**) ranging in method env-two-stage.

**Figure 24 sensors-22-07345-f024:**
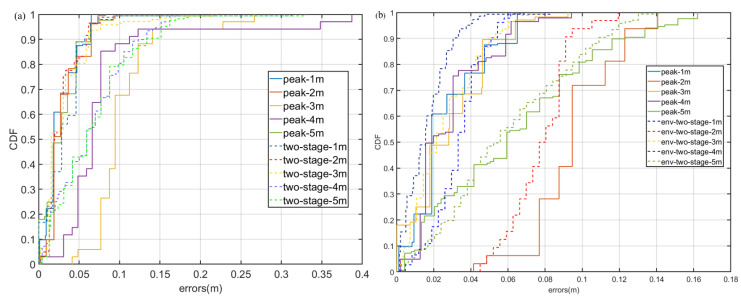
CDF in Case 1. (**a**) Comparison results between Peak and two-stage. (**b**) Comparison results between Peak and env-two-stage.

**Figure 25 sensors-22-07345-f025:**
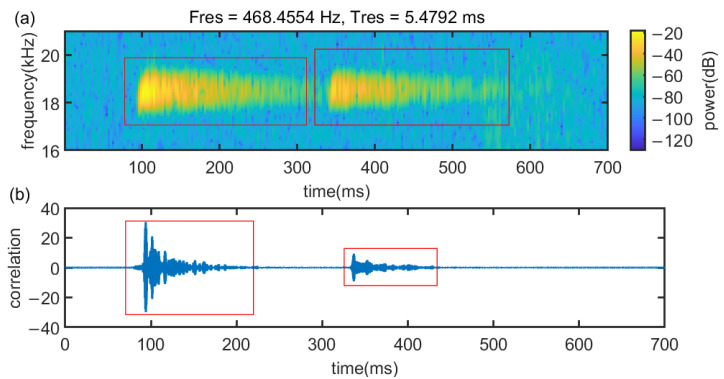
Signal propagation characteristics in case 2. (**a**) Signal spectrogram and (**b**) cross-correlation results.

**Figure 26 sensors-22-07345-f026:**
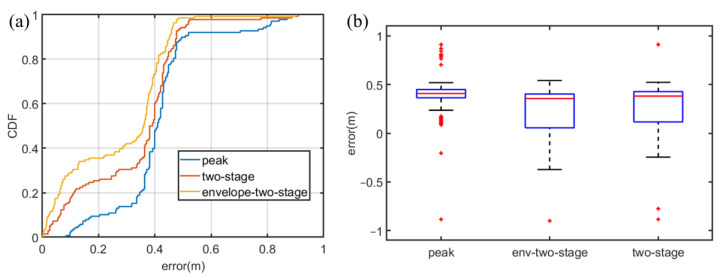
Experiment results in the situation of NLoS by people. (**a**) CDF of experimental error. (**b**) box figure of experimental error.

**Figure 27 sensors-22-07345-f027:**
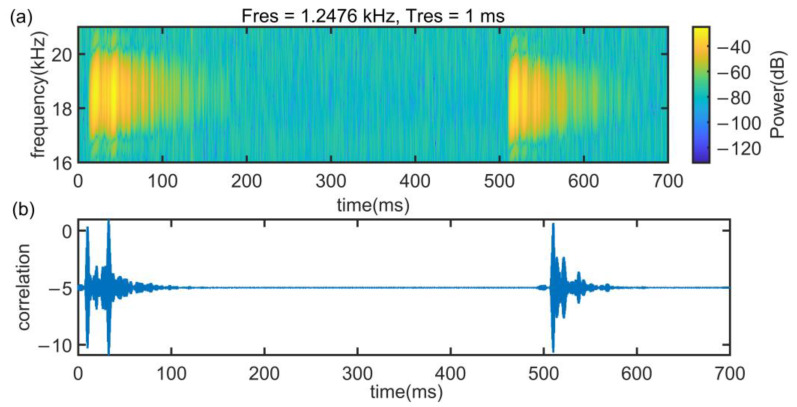
Signal propagation characteristics in case 3. (**a**) Signal Spectrum and (**b**) correlation results.

**Figure 28 sensors-22-07345-f028:**
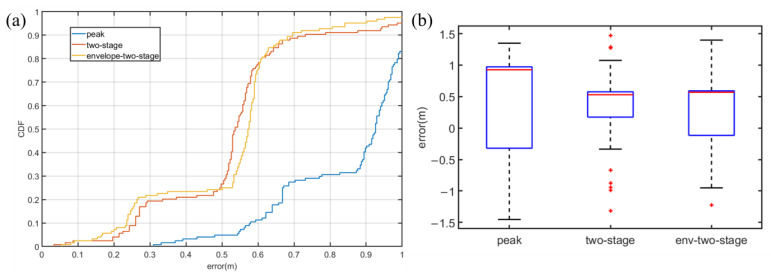
Experiment results in the canteen situation of NLoS. (**a**) box figure of experimental error. (**b**) CDF of experimental error.

**Figure 29 sensors-22-07345-f029:**
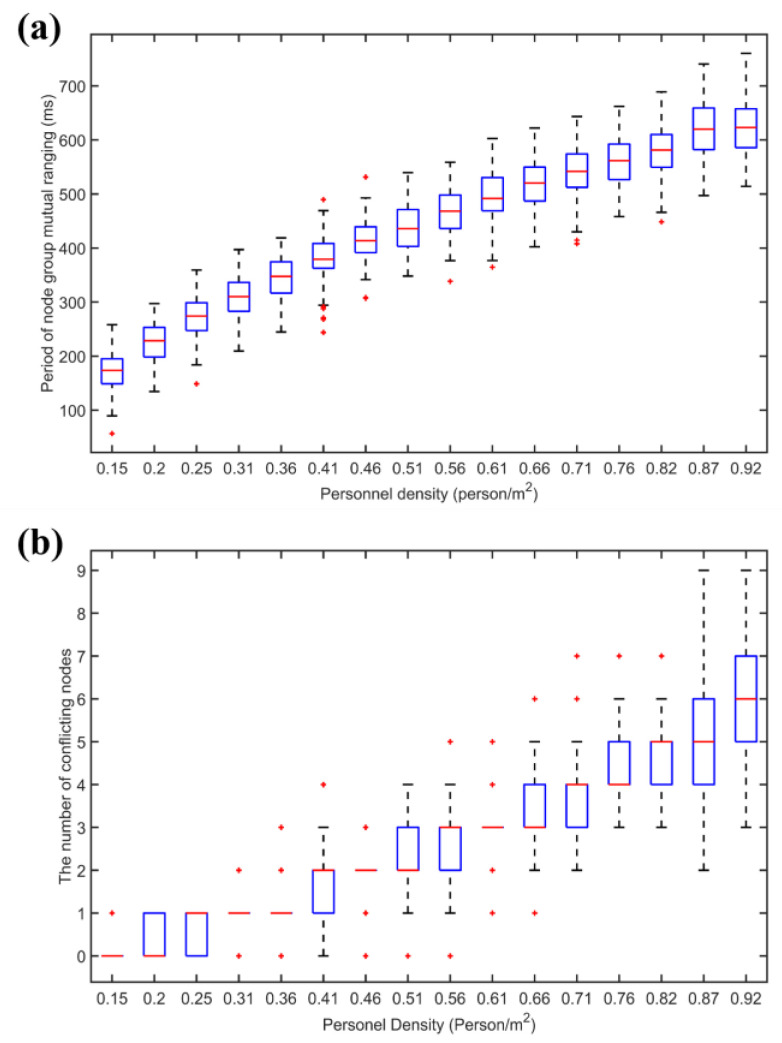
CSMA/CA channel access simulation results. (**a**) The total period of node cluster mutual ranging under hybrid channel access. (**b**) The number of nodes colliding with hybrid channel access.

**Figure 30 sensors-22-07345-f030:**
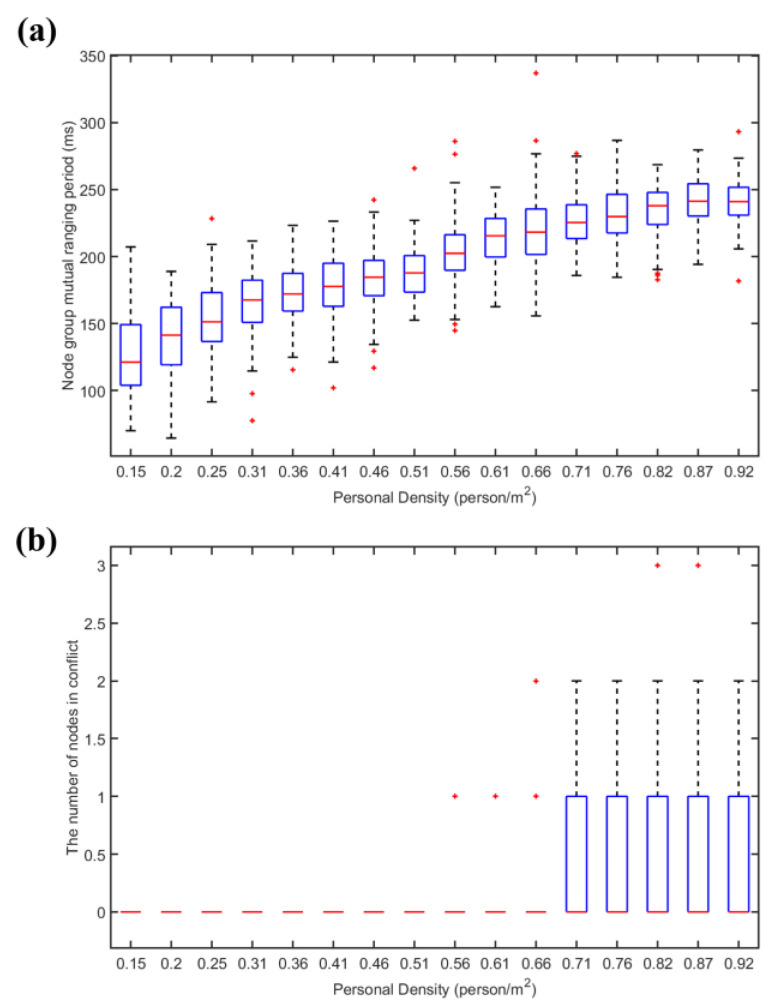
Hybrid channel access simulation results. (**a**) The total period of node cluster mutual ranging under hybrid channel access. (**b**) The number of nodes colliding with hybrid channel access.

**Figure 31 sensors-22-07345-f031:**
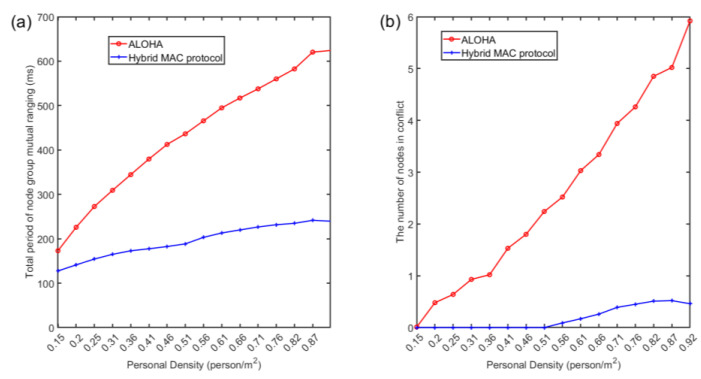
(**a**) The total period of mutual ranging and (**b**) the conflict situation of different channel access methods.

**Table 1 sensors-22-07345-t001:** The comparison of different techniques.

Technique	Accuracy (m)	Range (m)	Cost	Smartphone Compatibility	BS Requirement
Wi-Fi RTT	0.05~0.20	20	High	Compatible	Yes
Wi-Fi RSSI	3~8		Low	Compatible	Yes
BLE RSSI	2~5	10	Low	Compatible	Yes
BLE AOA	0.05~0.50		High	Compatible	Yes
BLE Ibeacon	0.05~0.10	1~2	Low	Compatible	No
UWB	0.05~0.30	50	High	Mostly Incompatible	Yes
PDR	1.62	/	Low	Compatible	No
Acoustic	0.1	40	Low	Compatible	No

**Table 2 sensors-22-07345-t002:** MAC simulation settings.

Parameter	Value
Diagram of border circle	5 m
Personal (node) number	3~18
Personal (node) density	0.15/m^2^~0.92/m^2^
Minimum personal distance	0.8 m
Maximum personal distance	5 m
Signal length	20 ms
$\mathrm{Maximum}\;\mathrm{time}\;\mathrm{delay}\;T_{m}$	14.58 ms
System time delay	5 ms
$T_{slot}$	$T_{m}+\mu_{s}+\mu_{r}$
IFS	$T_{m}+\mu_{s}+\mu_{r}$
Competition window	8
Simulation step	0.1 mm
Speed of sound	343 m/s
CSMA	P-resist (P = 0.8)

**Table 3 sensors-22-07345-t003:** Summary of all experimental results. Avd and Std are the average and stand error of the results.

Environment Setting	Method	Operational Range	Avd (m)	Std (m)	50% CDF (m)	95% CDF (m)
Case 1	Peak	1 m	0.0269	0.0190	0.0190	0.0650
Case 1	two-stage	1 m	0.0269	0.0304	0.0190	0.0650
Case 1	env-two-stage	1 m	0.0333	0.0133	0.0331	0.0544
Case 1	Peak	2 m	0.0948	0.0225	0.0946	0.2103
Case 1	two-stage	2 m	0.1038	0.0804	0.0662	0.3031
Case 1	env-two-stage	2 m	0.0776	0.0164	0.0786	0.1041
Case 1	Peak	3 m	0.0281	0.0214	0.0281	0.0640
Case 1	two-stage	3 m	0.0292	0.0315	0.0285	0.0640
Case 1	env-two-stage	3 m	0.0257	0.0180	0.0215	0.0604
Case 1	Peak	4 m	0.0279	0.0193	0.0198	0.0617
Case 1	two-stage	4 m	0.0283	0.0319	0.0156	0.0765
Case 1	env-two-stage	4 m	0.0163	0.0126	0.0156	0.0385
Case 1	Peak	5 m	0.0614	0.0426	0.0594	0.1515
Case 1	two-stage	5 m	0.0641	0.0668	0.0594	0.1525
Case 1	env-two-stage	5 m	0.0597	0.0351	0.0558	0.1196
Case 2	Peak	3 m	0.4113	0.1551	0.4096	0.8041
Case 2	two-stage	3 m	0.2316	0.1729	0.3866	0.5109
Case 2	env-two-stage	3 m	0.2060	0.1753	0.3600	0.4648
Case 3	Peak	1.5 m	0.8565	0.1946	0.9260	1.056
Case 3	two-stage	1.5 m	0.5618	0.2349	0.5364	1.009
Case 3	env-two-stage	1.5 m	0.5322	0.2140	0.5736	0.8591

## Data Availability

The data presented in this study are available on request from the corresponding author.

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
