# Peer review of "Smartphone-Based Social Distance Detection Technology with Near-Ultrasonic Signal"

_sensors, 2022, doi:10.3390/s22197345_

Round 1

Reviewer 1 Report

Dear Authors,

please consider the following remarks when finalizing the article:

1. It is necessary to provide a reference to the source of formula (3).

2. [43] does not seem to be a reference to an article about human hearing.

3. The notation in formula (8) (v, vr, vs) needs to be explained.

4. It is necessary to give a signature on the y-axis in Fig. 23

Reviewer 2 Report

The paper is an interesting piece of work on the topic of distance estimation based on wireless signals, here audio signals are used (inaudible for humans).

Strengths:

- The paper gives a very good overview of existing approaches and methods (BLE, UWB, etc.) and motivates the problem of inaccurate estimates well.

- The paper provides real experiments and presents the results clearly.

Weaknesses:

- It is not clear why new MAC algorithms are needed for the purpose of the work. If it is not really needed, I suggest to omit this aspect in this work.

- The experiment cases are limited to a few basic cases, yet in real setting, more complex constellations are expected.

Detailed comments:

- The cited works are relevant, yet I recommend to have a look at the following recent related work and include it in the related works section:

[1] B. Etzlinger, B. Nußbaummüller, P. Peterseil and K. A. Hummel, "Distance Estimation for BLE-based Contact Tracing – A Measurement Study," 2021 Wireless Days (WD), 2021, pp. 1-5, doi: 10.1109/WD52248.2021.9508280.

[2] D. J. Leith and S. Farrell, \Coronavirus Contact Tracing: Evaluating the Potential of Using
Bluetooth Received Signal Strength for Proximity Detection," SIGCOMM Comput. Commun.
Rev., vol. 50, no. 4, pp. 66-74, Oct. 2020. doi: 10.1145/3431832.3431840.

- page 4, UWB: It is true that few phones have UWB, but in the future, this may change. In [2] (see above), there is also a description about the potentials (and drawbacks) of UWB. I suggest to extend this section by a discussion of UWB as UWB is indeed a promising and relevant ranging technology.

- Section 3: Here, Microsoft BeepBeep is described, yet it is not clear which parts of this system are used and where own technology is applied. This needs to be clarified.

- Section 3.3: Figure 12 should be described in the text, i.e. what is provided by each of the steps (briefly).

- Experiments: How many runs have been conducted in each of the cases? Maybe I missed it but this should be clearly stated.

- Figure 23 and text: I assume that the error is given in meters (unit should be added to the figure). How is the error calculated? We see here a negative error, which is kind of misleading (e.g., assume two error values of 2 m and -2 m respectively, leading to a median/average error value of 0 m). I suggest to discuss the error in terms of a positive distance error.

Editorial:

- page2, line 84,85: "the system has little argument about the NLoS system" -- I suggest to reformulate and avoid "has little argument".

- Some figures and captions are not centered, I guess they should be in the CR version.

- I suggest also to use a space between the number and the unit Hz, e.g., page 11 "19250Hz" --> 19250 Hz.

Reviewer 3 Report

Overall contents and materials provided is good.  Paper Organization is also good.  Some suggestions are as under: 

 (Page 6 line no:  206)   “Optimazed Signal Designed”   title needs revision.

(Page 6 line no:  211):    it is reported that “Figure 5 shows the frequency responses of different commercial mobile phone microphones.”   Give experimentation details

(Page 6 line no:  215):  it is reported that “According to the results, this paper uses the acoustic signal frequency range of 18KHz-23KHz…”   which results are being referred? Give proper reasons or referencing.

 (Page 7 – line 230)  “The theory relationship between hearing threshold and frequency is shown as….” Give citation or Justify reasons for parameters’ selection for equation (3).

(Page 8 – line 253)   In order to solve low-frequency leakage problem, the waveform of the chirp signal is reconstructed, and the window of Blackman window and rectangular window is used. Justify the selection of window(s).
